# ASYMPTOTIC LEARNING CURVES OF KERNEL METHODS: EMPIRICAL DATA *v.s.* TEACHER-STUDENT PARADIGM

## ABSTRACT

How many training data are needed to learn a supervised task? It is often observed that the generalization error decreases as $n^{-\beta}$ where $n$ is the number of training examples and $\beta$ an exponent that depends on both data and algorithm. In this work we measure $\beta$ when applying kernel methods to real datasets. For MNIST we find $\beta \approx 0.4$ and for CIFAR10 $\beta \approx 0.1$. Remarkably, $\beta$ is the same for regression and classification tasks, and for Gaussian or Laplace kernels. To rationalize the existence of non-trivial exponents that can be independent of the specific kernel used, we introduce the Teacher-Student framework for kernels. In this scheme, a Teacher generates data according to a Gaussian random field, and a Student learns them via kernel regression. With a simplifying assumption — namely that the data are sampled from a regular lattice — we derive analytically $\beta$ for translation invariant kernels, using previous results from the kriging literature. Provided that the Student is not too sensitive to high frequencies, $\beta$ depends only on the training data and their dimension. We confirm numerically that these predictions hold when the training points are sampled at random on a hypersphere. Overall, our results quantify how smooth Gaussian data should be to avoid the curse of dimensionality, and indicate that for kernel learning the relevant dimension of the data should be defined in terms of how the distance between nearest data points depends on $n$. With this definition one obtains reasonable effective smoothness estimates for MNIST and CIFAR10.

## 1 INTRODUCTION

In supervised learning machines learn from a finite collection of $n$ training data, and their generalization error is then evaluated on unseen data drawn from the same distribution. How many data are needed to learn a task is characterized by the learning curve relating generalization error to $n$. In various cases, the generalization error decays as a power law $n^{-\beta}$, with an exponent $\beta$ that depends on both the data and the algorithm. In (Hestness et al., 2017) $\beta$ is reported for state-of-the-art (SOTA) deep neural networks for various tasks: in for *neural-machine translation* $\beta \approx 0.3$–$0.36$ (for fixed model size) or $\beta \approx 0.13$ (for best-fit models at any $n$); *language modeling* shows $\beta \approx 0.06$–$0.09$; in *speech recognition* $\beta \approx 0.3$; SOTA models for *image classification* (on ImageNet) have exponents $\beta \approx 0.3$–$0.5$. Currently there is no available theory of deep learning to rationalize these observations. Recently it was shown that for a proper initialization of the weights, deep learning in the infinite-width limit (Jacot et al., 2018) converges to kernel learning. Moreover, it is nowadays part of the lore that there exist kernels whose performance is nearly comparable to deep networks (Bruna and Mallat, 2013; Arora et al., 2019), at least for some tasks. It is thus of great interest to understand the learning curves of kernels. For regression, if the target function being learned is simply assumed to be Lipschitz, then the best guarantee is $\beta = 1/d$ (Luxburg and Bousquet, 2004; Bach, 2017) where $d$ is the data dimension. Thus for large $d$, $\beta$ is very small: learning is completely inefficient, a phenomenon referred to as the *curse of dimensionality*. As a result, various works on kernel regression make the much stronger assumption that the training points are sampled from a target function that belongs to the *reproducing kernel Hilbert space* (RKHS) of the kernel (see for example (Smola et al., 1998)). With this assumption $\beta$ does not depend on $d$ (for instance in (Rudi and Rosasco, 2017) $\beta = 1/2$ is guaranteed). Yet, RKHS is a very strong assumption which requires the smoothness of the target

function to increase with $d$ (Bach, 2017) (see more on this point below), which may not be realistic in large dimensions.

In this work we compute $\beta$ empirically for kernel methods applied on MNIST and CIFAR10 datasets. We find $\beta_{\text{MNIST}} \approx 0.4$ and $\beta_{\text{CIFAR10}} \approx 0.1$ respectively. Quite remarkably, we observe essentially the same exponents for regression and classification tasks, using either a Gaussian or a Laplace kernel. Thus the exponents are not as small as $1/d$ ($d = 784$ for MNIST, $d = 3072$ for CIFAR10), but neither are they $1/2$ as one would expect under the RKHS assumption. These facts call for frameworks in which assumptions on the smoothness of the data can be intermediary between Lipschitz and RKHS. Here we propose such a framework for regression, in which the target function is assumed to be a Gaussian random field of zero mean with translation-invariant isotropic covariance $K_T(\underline{x})$. The data can equivalently be thought as being synthesized by a "Teacher" kernel $K_T(\underline{x})$. Learning is performed with a "Student" kernel $K_S(\underline{x})$ that minimizes the mean-square error. In general $K_T(\underline{x}) \neq K_S(\underline{x})$. In this set-up learning is very similar to a technique referred to as *kriging*, or Gaussian process regression, originally developed in the geostatistics community (Matheron, 1963; Stein, 1999b). To quantify learning, we first perform numerical experiments for data points distributed uniformly at random on a hypersphere of varying dimension $d$, focusing on a Laplace kernel for the Student, and considering a Laplace or Gaussian kernel for the Teacher. We observe that in both cases $\beta(d)$ is a decreasing function.

To derive $\beta(d)$ we consider the simplified situation where the Gaussian random field is sampled at training points lying on a regular lattice. Building on the kriging literature (Stein, 1999b), we show that $\beta$ is controlled by the high-frequency scaling of both the Teacher and Student kernels: assuming that the Fourier transforms of the kernels decay as $\tilde{K}_T(\underline{w}) = c_T\|\underline{w}\|^{-\alpha_T} + o\left(\|\underline{w}\|^{-\alpha_T}\right)$ and $\tilde{K}_S(\underline{w}) = c_S\|\underline{w}\|^{-\alpha_S} + o\left(\|\underline{w}\|^{-\alpha_S}\right)$, we obtain

$$\beta = \frac{1}{d}\min(\alpha_T - d, 2\alpha_S). \tag{1}$$

Importantly (i) Eq. (1) leads to a prediction for $\beta(d)$ that accurately matches our numerical study for random training data points, leading to the conjecture that Eq. (1) holds in that case as well. We offer the following interpretation: ultimately, kernel methods are performing a local interpolation whose quality depends on the distance $\delta(n)$ between adjacent data points. $\delta(n)$ is asymptotically similar for random data or data sitting on a lattice. (ii) If the kernel $K_S$ is not too sensitive to high-frequencies, then learning is optimal as far as scaling is concerned and $\beta = (\alpha_T - d)/d$. We will argue that the smoothness index $s \equiv [(\alpha_T - d)/2]$ characterizes the number of derivatives of the target function that are continuous. We thus recover the curse of dimensionality: $s$ needs to be of order $d$ to have non-vanishing $\beta$ in large dimensions. Point (ii) leads to an apparent paradox: $\beta$ is significant for MNIST and CIFAR10, for which $d$ is a priori very large, leading to a smoothness value $s$ in the hundreds in both cases, which appears unrealistic. The paradox is resolved by considering that real datasets actually live on lower-dimensional manifolds. As far as kernel learning is concerned, our findings support that the correct definition of dimension should be based on how the nearest-neighbors distance $\delta(n)$ scales with $n$: $\delta(n) \sim n^{-1/d_{\text{eff}}}$. Direct measurements of $\delta(n)$ support that MNIST and CIFAR10 live on manifolds of lower dimensions $d_{\text{MNIST}}^{\text{eff}} \approx 15$ and $d_{\text{CIFAR10}}^{\text{eff}} \approx 35$. Considering the effective dimensions that we find, the observed values for $\beta$ would be obtained for Gaussian fields of smoothness $s_{\text{MNIST}} \approx 3$ and $s_{\text{CIFAR10}} \approx 1$, values that appear intuitively more reasonable. More generally this analogy with Gaussian fields allows one to associate a smoothness index $s$ to any dataset once $\beta$ and $d_{\text{eff}}$ are measured, which may turn out to be a useful characterization of data complexity in the future.

## 2 RELATED WORKS

Our set-up of Teacher-Student learning with kernels is also referred to as *kriging*, or Gaussian process regression, and it was originally developed in the geostatistics community (Matheron, 1963). In Section 5 we present a theorem that allows one to know the rate at which the test error decreases as we increase the number of training points, assumed to lie on a high-dimensional regular lattice. Similar results have been previously derived in the kriging literature (Stein, 1999b) when sampling occurs on the regular lattice with the exception of the origin, where the inference is made. Here we propose an alternative derivation that some readers might find simpler. We also study a slightly different problem: instead of computing the test error when the inference is carried on at the origin,

we compute the average error for a test point that lie at an arbitrary point, sampled uniformly at random and not necessarily on the lattice.

In what follows we show, via extensive numerical simulations, that such predictions are accurate even when the training points do not lie on a regular lattice, but are taken at random on a hypersphere. An exact proof of our result in such a general setting is difficult and cannot be found even in the kriging literature. To our knowledge the results that get closer to the point are those discussed in (Stein, 1999a), where the author studies one-dimensional processes where the training data are not necessarily evenly spaced.

In this work the effective dimension of the data plays an import role, as it controls how the distance between nearest neighbors scales with the dataset size. Of course, there exists a vast literature (Grassberger and Procaccia, 1983; Costa and Hero, 2004; Hein and Audibert, 2005; Levina and Bickel, 2005; Rozza et al., 2012; Facco et al., 2017; Allegra et al., 2019) devoted to the study of effective dimensions, where other definitions are analyzed. The effective dimensions that we find are compatible with those obtained with more refined methods.

## 3 LEARNING CURVE FOR KERNEL METHODS APPLIED TO REAL DATA

In what follows we apply kernel methods to the MNIST and CIFAR10 datasets, each consisting of a set of images $(\underline{x}_\mu)_{\mu=1}^n$. We simplify the problem by considering only two classes whose label $Z(\underline{x}_\mu) = \pm 1$ correspond to odd and even numbers for MNIST, and to two groups of 5 classes in CIFAR10. The goal is to infer the value of the label $\hat{Z}_S(\underline{x})$ of an image $\underline{x}$ that does not belong to the dataset. The $S$ subscript reminds us that inference is performed using a positive definite kernel $K_S$. We perform inference in both a *regression* and a *classification* setting. The following algorithms and associated results can be found in (Scholkopf and Smola, 2001).

**Regression.**   Learning corresponds to minimizing a mean-square error:

$$\min \sum_{\mu=1}^n \left[ \hat{Z}_S(\underline{x}_\mu) - Z(\underline{x}_\mu) \right]^2 . \tag{2}$$

For algorithms seeking solutions of the form $\hat{Z}_S(\underline{x}) = \sum_\mu a_\mu K_S(\underline{x}_\mu, \underline{x}) \equiv \underline{a} \cdot \underline{k}_S(\underline{x})$ by minimizing the man-square loss over the vector $\underline{a}$, one obtains:

$$\hat{Z}_S(\underline{x}) = \underline{k}_S(\underline{x}) \cdot \mathbb{K}_S^{-1} \underline{Z}, \tag{3}$$

where the vector $\underline{Z}$ contains all the labels in the training set, $\underline{Z} \equiv (Z(\underline{x}_\mu))_{\mu=1}^n$, and $\mathbb{K}_{S,\mu\nu} \equiv K_S(\underline{x}_\mu, \underline{x}_\nu)$ is the Gram matrix. The Gram matrix is always invertible if the kernel $K_S$ is positive definite. The generalization error is then evaluated as the expected mean-square error on unseen data, estimated by averaging over a test set composed of $n_{\text{test}}$ unseen data points:

$$\text{MSE} = \frac{1}{n_{\text{test}}} \sum_{\mu=1}^{n_{\text{test}}} \left[ \hat{Z}_S(\underline{x}_\mu) - Z(\underline{x}_\mu) \right]^2 . \tag{4}$$

**Classification.**   We perform kernel classification via the algorithm *soft-margin SVM*. The details can be found in Appendix A. After learning from the training data with a student kernel $K_S$, performance is evaluated via the generalization error. It is estimated as the fraction of correctly predicted labels for data points belonging to a test set with $n_{\text{test}}$ elements.

In Fig. 1 we present the learning curves for (binary) MNIST and CIFAR10, for regression and classification. Learning is performed both with a Gaussian kernel $K(\underline{x}) \propto \exp(-\|\underline{x}\|^2/(2\sigma^2))$ and a Laplace one $K(\underline{x}) \propto \exp(-\|\underline{x}\|/\sigma)$. Remarkably, the power laws in the two tasks are essentially identical (although the estimated exponent appears to be slightly larger, in absolute value, for classification). Moreover, the two kernels display a very similar behavior, compatible with the same exponent: about $-0.4$ for MNIST and $-0.1$ for CIFAR10. The presented data are for $\sigma = 1000$; in Appendix B we show that the same behaviour is observed for different values.

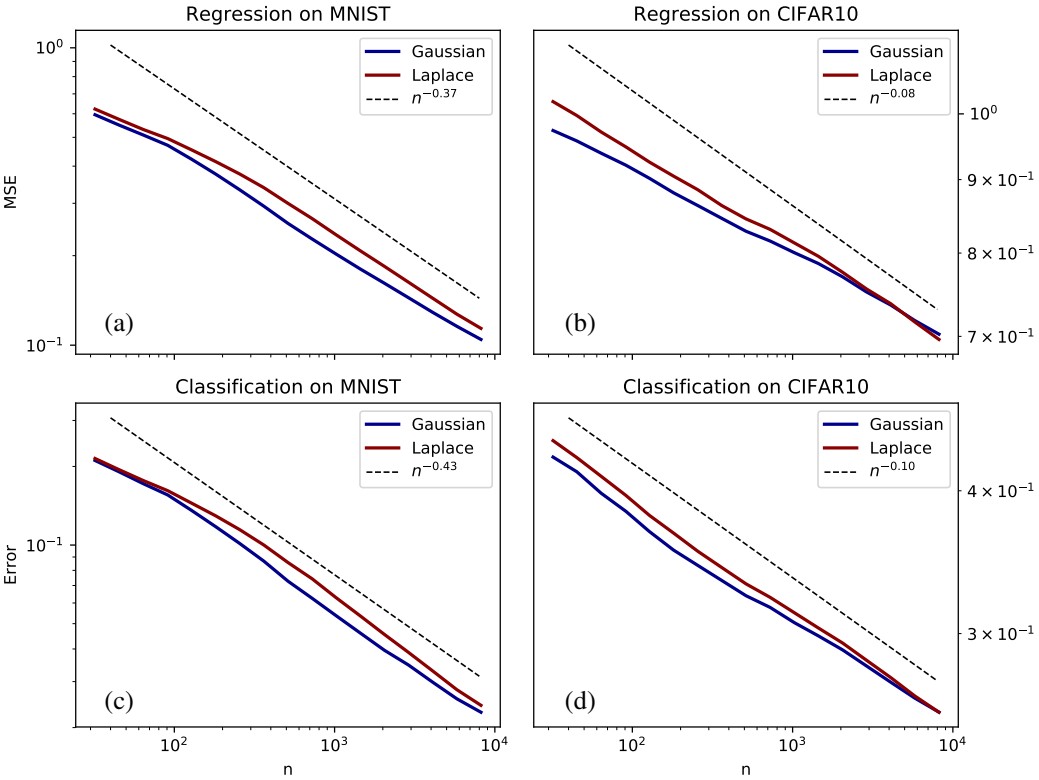

Figure 1: Learning curves for regression on MNIST and CIFAR10 *(a-b)*; and for classification on MNIST and CIFAR10 *(c-d)*. Curves are averaged over $400$ runs. A power law is plotted to estimate the asymptotic behavior at large $n$: the exponent is fitted on the last decade on the average of the two curves, since it does not seem to depend significantly on the specific kernel or on the task. In each setting we use both a Gaussian kernel $K(\underline{x}) \propto \exp(-\|\underline{x}\|^2/(2\sigma^2))$ and a Laplace one $K(\underline{x}) \propto \exp(-\|\underline{x}\|/\sigma)$, with $\sigma = 1000$.

## 4 GENERALIZATION SCALING IN KERNEL TEACHER-STUDENT PROBLEMS

We study $\beta$ in a simplified setting where the data is assumed to follow a Gaussian distribution with known covariance. It falls into the class of teacher-Student problems, which are characterized by a machine (the Teacher) that generates the data, and another machine (the Student) that tries to learn from them. The Teacher-Student paradigm has been broadly used to study supervised learning (Saad and Solla, 1995; Monasson and Zecchina, 1995; Opper and Saad, 2001; Engel and Van den Broeck, 2001; Zdeborová and Krzakala, 2016; Barbier et al., 2019; Gabrié et al., 2018; Aubin et al., 2018; Franz et al., 2018). He we restrict our attention to kernel methods: we assume that a target function is distributed according to a Gaussian random field $Z \sim \mathcal{N}(0, K_T)$ — the Teacher — characterized by a translation-invariant isotropic covariance function $K_T(\underline{x}, \underline{x}') = K_T(\|\underline{x} - \underline{x}'\|)$, and that the training dataset consists the finite set of $n$ observations $\underline{Z} = (Z(\underline{x}_\mu))_{\mu=1}^n$. This is equivalent to saying that the vector of training points follows a centered Gaussian distribution with a covariance matrix that depends on $K_T$ and on the location of the points $(\underline{x}_\mu)_{\mu=1}^n$:

$$\underline{Z} \sim \mathcal{N}\left(\underline{0}, \mathbb{K}_T\right), \quad \text{where} \quad \mathbb{K}_T = \left(K_T(\underline{x}_\mu, \underline{x}_\nu)\right)_{\mu,\nu=1}^n. \tag{5}$$

Once the Teacher has generated the dataset, the rest follows as in the kernel regression described in the previous section. We use another translation-invariant isotropic kernel $K_S(\underline{x}, \underline{x}')$ — the Student — to infer the value of the field at another point, $\hat{Z}_S(\underline{x})$, with a regression task, i.e. minimizing the mean-square error in Eq. (2). The solution is therefore given again by Eq. (3).

Fig. 2 *(a-b)* shows the mean-square error obtained numerically. In the examples the Student is always taken to be a Laplace kernel, and the Teacher is either a Laplace kernel or a Gaussian kernel. The

points $(\underline{x}_\mu)_{\mu=1}^n$ are taken uniformly at random on the unit $d$-dimensional hypersphere for several dimensions $d$ and for several dataset sizes $n$. We take $\sigma_S = \sigma_T = d$ as we observed that with this choice smaller datasets were enough to approach a limiting curve — in Appendix C we show the plots for the case $\sigma_S = \sigma_T = 10$, which appears to converge to the same limit curve with increasing $n$, but at a smaller pace. The figure shows that when $n$ is large enough, the mean-square error behaves as a power law (dashed lines) with an exponent that depends on the spatial dimension of the data, as well as on the kernels. The fitted exponents are plotted in Fig. 2 *(c-d)* as a function of the spatial dimension $d$ for different dataset sizes $n$. In the next section we will discuss the theoretical prediction, that in the figure is plotted a thick black line. The figure shows that as the dataset gets bigger, the asymptotic exponent tends to our prediction. In Appendix D we present the learning curves of Gaussian Students with both a Laplace and a Gaussian kernel. When both kernels are Gaussian the test error decays exponentially fast, a result that matches our theoretical prediction. In Appendix E we also provide further numerical results for the case where the Teacher kernel is a Matérn kernel (as defined therein).

## 5 ANALYTIC ASYMPTOTICS FOR THE KERNEL TEACHER-STUDENT PROBLEM ON A LATTICE

In this section we compute analytically the exponent that describe the asymptotic decay of the generalization error when the number $n$ of training data increases. In order to derive the result we assume that both the Teacher Gaussian random field lives on a bounded hypercube, $\underline{x} \in \mathcal{V} \equiv [0, L]^d$, where $L$ is a constant and $d$ is the spatial dimension. The fields and the kernels can then be thought of

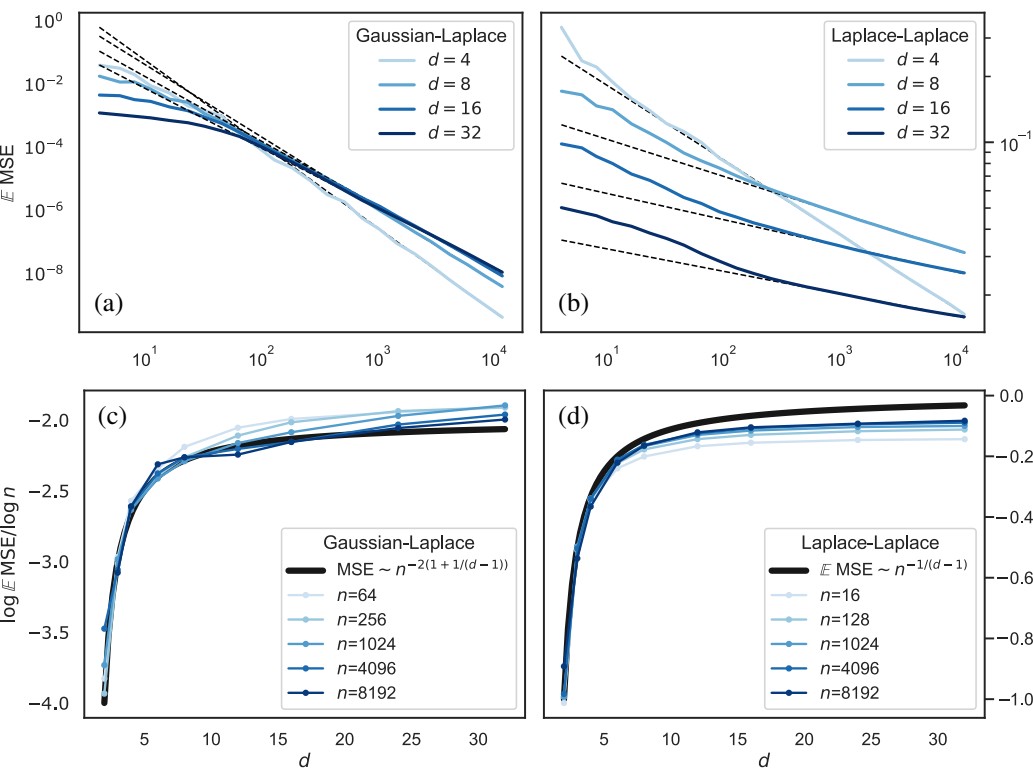

Figure 2: Results for the Teacher-Student kernel regression problem, where the Student is always a Laplace kernel. Data points are sampled uniformly at random on a $d$-dimensional hypersphere. *(a-b)* Mean-square error versus the size of the training dataset, for Gaussian and Laplace Teachers and for multiple spatial dimensions. Dotted lines are the fitted power laws — we fit starting from $n = 700$. *(c-d)* Fitted exponent $-\beta = \log \mathbb{E} \, \mathrm{MSE} / \log n$ against the spatial dimension, for several dataset sizes. We fit from $n = 0$ to a varying $n$ (written in the legends). The thick black lines are the theoretical predictions.

as $L$-periodic along each dimension. Furthermore, to make the problem tractable we assume that the points $(\underline{x}_\mu)_{\mu=1}^n$ live on a *regular lattice*, covering all the hypercube $\mathcal{V}$. Therefore, the linear spacing between neighboring points is $\delta = Ln^{-1/d}$. This is of course a different setting than the one used in the numerical simulations presented in the previous section, yet our results below support that these differences do not matter.

Generalization error is then evaluated via the typical mean-square error

$$\mathbb{E}\,\text{MSE} = \mathbb{E}\left[ Z(\underline{x}) - \hat{Z}_S(\underline{x}) \right]^2 , \tag{6}$$

where the expectation is taken over both the Teacher process and the point $\underline{x}$ at which we estimate the field, assumed to be uniformly distributed in the hypercube $\mathcal{V}$. In Appendix F we prove the following:

---

**Theorem 1.** *Let $\tilde{K}_T(\underline{w}) = c_T\|\underline{w}\|^{-\alpha_T} + o\left(\|\underline{w}\|^{-\alpha_T}\right)$ and $\tilde{K}_S(\underline{w}) = c_S\|\underline{w}\|^{-\alpha_S} + o\left(\|\underline{w}\|^{-\alpha_S}\right)$ as $\|\underline{w}\| \to \infty$, where $\tilde{K}_T(\underline{w})$ and $\tilde{K}_S(\underline{w})$ are the Fourier transforms of the kernels $K_T(\underline{x})$, $K_S(\underline{x})$ respectively, assumed to be positive definite. We assume $\tilde{K}_T(\underline{w})$ and $\tilde{K}_S(\underline{w})$ has a finite limit as $\|\underline{w}\| \to 0$ and that $K(\underline{0}) < \infty$. Then,*

$$\mathbb{E}\,\text{MSE} = n^{-\beta} + o\left(n^{-\beta}\right) \quad \text{with} \quad \beta = \frac{1}{d}\min(\alpha_T - d, 2\alpha_S). \tag{7}$$

*Moreover, in the case of a Gaussian kernel the result holds valid if we take the corresponding exponent to be $\alpha = \infty$.*

---

Apart from the specific value of the exponent in Eq. (7), Theorem 1 implies that if the Student kernel decays fast enough in the frequency domain, then $\beta$ depends only on the data through the behaviour of the Teacher kernel at high frequencies. One then recovers $\beta = (\alpha_T - d)/d$, also found for the *Bayes-optimal* setting where the Student is identical to the Teacher.

Consider the predictions of Theorem 1 in the cases presented in Fig. 2 *(a-b)* of Gaussian and Laplace kernels. If both kernels are Laplace kernels then $\alpha_T = \alpha_S = d + 1$ and $\mathbb{E}\,\text{MSE} \sim n^{-1/d}$, which scales very slowly with the dataset size in large dimensions. If the Teacher is a Gaussian kernel ($\alpha_T = \infty$) and the Student is a Laplace kernel then $\beta = 2(1 + 1/d)$, leading to $\beta \to 2$ as $d \to \infty$. In Fig. 2 *(c-d)* we compare these predictions with the exponents extracted from Fig. 2 *(a-b)*. We plot $\log \mathbb{E}\,\text{MSE}/\log n \equiv -\beta$, against the dimension $d$ of the data, varying the dataset size $n$. The exponents extracted numerically tend to our analytical predictions when $n$ is large enough.

Notice that, although the theory and the experiments do not assume the same distribution for the sampling points $(\underline{x}_\mu)_{\mu=1}^n$, this does not seem to yield any difference in the asymptotic behavior of the generalization error, leading to the conjecture that our predictions are exact even when the training set is random, and does not correspond to a lattice. The conjecture can be proven in one dimension following results of the kriging literature (Stein, 1999a), but generalization to higher $d$ is a much harder problem. Intuitively, for kernel learning performs an expansion, whose quality is governed by the target function smoothness and the typical distance $\delta_{\min}$ between a point and its nearest neighbors in the training set. Both for random points or on a lattice, one has $\delta_{\min} \sim n^{-1/d}$ when $n$ is large enough, thus both situations lead to the same $\beta$.

Theorem 1 underlines that kernel methods are subjected to the curse of dimensionality. Indeed for appropriate students, one obtains $\beta = (\alpha_T - d)/d$. Let us define the smoothness index $s \equiv [(\alpha_T - d)/2] = \beta d/2$, which must be $\mathcal{O}(d)$ to avoid $\beta \to 0$ for large $d$. The two Lemmas below, derived in Appendix, indicate that the target function is $s$ time differentiable (in a mean-square sense). Thus learning with kernels in very large dimension can only occur if the target function is $\mathcal{O}(d)$ times differentiable, a condition that appears very restrictive in large $d$.

---

**Lemma 1.** *Let $K(\underline{x}, \underline{x}')$ be a translation-invariant isotropic kernel such that $\tilde{K}(\underline{w}) = c\|\underline{w}\|^{-\alpha} + o\left(\|\underline{w}\|^{-\alpha}\right)$ as $\|\underline{w}\| \to \infty$ and $\|\underline{w}\|^d \tilde{K}(\underline{w}) \to 0$ as $\|\underline{w}\| \to 0$. If $\alpha > d + n$ for some $n \in \mathbb{Z}^+$, then $K(\underline{x}) \in C^n$, that is, it is at least $n$-times differentiable. (Proof in Appendix G).*

---

---

**Lemma 2.** *Let $Z \sim \mathcal{N}(0, K)$ be a d-dimensional Gaussian random field, with $K \in C^{2n}$ being a 2n-times differentiable kernel. Then Z is n-times* mean-square *differentiable in the sense that*

- *derivatives of $Z(\underline{x})$ are a Gaussian random fields;*

- $\mathbb{E}\partial_{x_1}^{n_1} \cdots \partial_{x_d}^{n_d} Z(\underline{x}) = 0$;

- $\mathbb{E}\partial_{x_1}^{n_1} \cdots \partial_{x_d}^{n_d} Z(\underline{x}) \cdot \partial_{x_1}^{n'_1} \cdots \partial_{x_d}^{n'_d} Z(\underline{x}') = \partial_{x_1}^{n_1+n'_1} \cdots \partial_{x_d}^{n_d+n'_d} K(\underline{x} - \underline{x}') < \infty$ *if the derivatives of K exist.*

*In particular, $\mathbb{E}\partial_{x_i}^m Z(\underline{x}) \cdot \partial_{x_i}^m Z(\underline{x}') = \partial_{x_i}^{2m} K(\underline{x} - \underline{x}') < \infty \ \forall m \leq n$. (Proof in Appendix G).*

---

## 6    EFFECTIVE DIMENSION OF DATA SETS

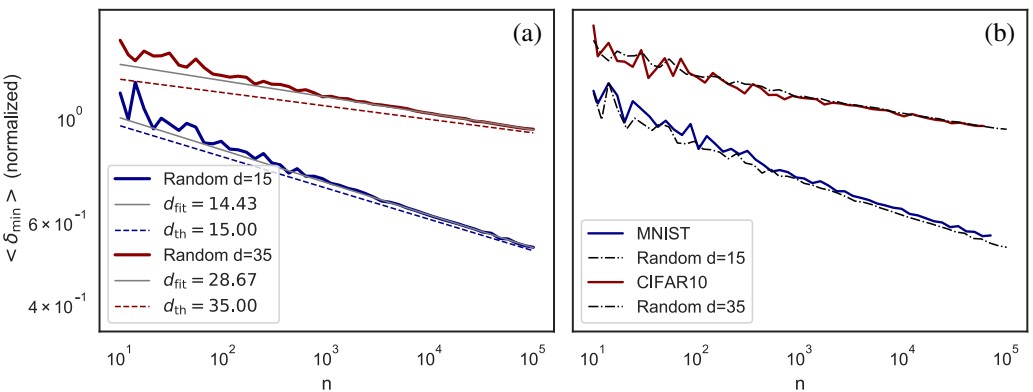

Figure 3: Average distance from one point to its nearest neighbor as a function of the dataset size $n$. *(a)* For random points on $d$-dimensional hypersphere, $\langle\delta_{\min}\rangle \sim n^{-1/d}$. Colored solid curves are found numerically, dashed lines are the theoretical asymptotic prediction and the gray lines are numerical fit (we fitted only starting from $n \approx 6000$ to reduce finite size effects, and the fit have been rescaled to match the data at $n = 10$). The larger $d$, the stronger the preasymptotic effects (a larger $n$ is needed to observe the predicted scaling). *(b)* Comparison between random data on 15- and 35-dimensional hyperspheres and the MNIST, CIFAR10 datasets. According to this definition of effective dimension, MNIST live on a 15-dimensional manifold and CIFAR10 on a 35-dimensional one. Data have been rescaled along the $y$-axis for ease of comparison.

If we approximate the high-dimensional MNIST and CIFAR10 datasets with Gaussian random fields, to obtain the curves shown in Fig. 1 and to find the values the we report for $\beta$ these fields would have to be hundreds of times differentiable, which seems unrealistic. A possible resolution of this paradox lies in the fact that the data live in a much smaller manifold than the number of pixels of these pictures would suggest. As argued above, a key determinant of kernel performance is the typical distance $\delta_{\min}$ between a point in the training set and its nearest neighbor. We define the effective dimension $d_{\text{eff}}$ accordingly from the asymptotic relationship between $\delta_{\min}$ and $n$:

$$\delta_{\min} \sim n^{-1/d_{\text{eff}}}. \tag{8}$$

For random points on a $d$-dimensional hypersphere $\delta_{\min}$ displays fluctuations and the scaling is valid only on average and only asymptotically, that is for $n$ larger than some characteristic scale $n^{\star}(d)$ that depends on the spatial dimension. In Fig. 3 *(a)* we show how the typical $\delta_{\min}$ scales with the dataset size $n$ for random points on hyperspheres of dimension $d = 15$ and $d = 35$. Notice that while for $d = 15$ the asymptotic regime is reached when $n \gtrsim 10^4$, for $d = 35$ a larger dataset is needed, with $n > 10^5$ points (that is about the maximum size of the dataset that we can use to apply kernel methods in our simulations, due to memory constraints). One can naturally wonder whether real data are also subjected to a scaling relation like in Eq. (8), from which an effective dimension can be defined. Consider for instance the MNIST dataset, and sample from it a subset of $n$ pictures. For each

point we could compute the distance from its nearest neighbor and average such quantities. As the number of data points increases, we expect that this measure characterizes the geometry of the local manifold where the data live in, since nearest neighbors are going to be closer and closer. In Fig. 3 *(b)* we present how $\delta_{\min}$ scales with $n$ for the MNIST and CIFAR10 datasets. Both display a power-law decay, but the exponent is not compatible with $1/d$ with $d$ the spatial dimension, namely $d = 784$ for MNIST and $d = 3072$ for CIFAR10. MNIST actually seems to scale pretty much like random data on a hypersphere with $d = 15$, and CIFAR10 scales approximately as random data on a hypersphere with $d = 35$. For this reason, the *effective dimensions* of these datasets are consistent with:

$$d_{\text{MNIST}}^{\text{eff}} \approx 15$$

$$d_{\text{CIFAR10}}^{\text{eff}} \approx 35.$$

Obviously, the intrinsic dimension of the data could vary in data space, as has been reported for MNIST (Costa and Hero, 2004; Hein and Audibert, 2005; Rozza et al., 2012; Facco et al., 2017). In this qualitative discussion we neglect such subtle effects. Interestingly, our effective dimensions leads to reasonable values for the effective smoothness:

$$s_{\text{eff}} = \beta d_{\text{eff}}/2.$$

In particular we find $s \approx 3$ for MNIST and $s \approx 1$ for CIFAR10.

## 7 CONCLUSION

In this work we have shown for CIFAR10 and MNIST respectively that kernel regression and classification display a power-law decay in the learning curves, quite remarkably with essentially the same exponent $\beta$, found to be larger for MNIST. These exponents are much larger than $\beta = 1/d$ expected for Lipschitz target functions and smaller than $\beta = 1/2$ expected for RKHS target functions. This observation led us to introduce a framework in which data are modeled as Gaussian random fields of varying smoothness, in which intermediary values of $\beta$ are obtained.

It is important to note the high degree of smoothness underlying the RKHS hypothesis. Consider realizations $Z(\underline{x})$ of a Teacher Gaussian process with covariance $K_T$ and assume that they lie in the RKHS of the Student kernel $K_S$, namely

$$\mathbb{E}\|Z\|_{K_S}^2 = \mathbb{E} \int \mathrm{d}\underline{x}\mathrm{d}\underline{y}Z(\underline{x})K_S^{-1}(\underline{x} - \underline{y})Z(\underline{y}) = \int \mathrm{d}\underline{w}\, \tilde{K}_T(\underline{w})\tilde{K}_S^{-1}(\underline{w}) < \infty. \tag{9}$$

If the Teacher and Student kernels decay in the frequency domain with exponents $\alpha_T$ and $\alpha_S$ respectively, convergence requires $\alpha_T > \alpha_S + d$, and $K_S(\underline{0}) \propto \int \mathrm{d}\underline{w}\, \tilde{K}_S(\underline{w}) < \infty$ (true for many commonly used kernels) implies $\alpha_S > d$. Then using Lemma 1 and Lemma 2 we can conclude that the realizations $Z(\underline{x})$ must be at least $\lfloor d/2 \rfloor$-times mean-square differentiable to be RKHS.

From this perspective, the RKHS assumption appears to be very strong, and thus may not provide an accurate description of various empirical learning curves. Our assumption that data are generated by Gaussian random processes is milder, and may thus have broader applications. Yet, we view this approximation as a first step on which to build on, to later include other effects such as noise in the data and deviations from Gaussianity.

ACKNOWLEDGMENTS

*Anonymized for double-blind review.*

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

## A  SOFT-MARGIN SUPPORT VECTOR MACHINES

The kernel classification task is performed via the algorithm known as *soft-margin Support Vector Machine*.

We want to find a function $\hat{Z}_S(\underline{x})$ such that its sign correctly predicts the label of the data. In this context we model such a function as a linear prediction after projecting the data on a *feature space* via $\underline{x} \to \phi(\underline{x})$:

$$\hat{Z}_S(\underline{x}) = \underline{w} \cdot \underline{\phi}_S(\underline{x}_\mu) + b, \tag{10}$$

where $\underline{w}, b$ are parameters to be learned from the training data. The kernel is related to the feature space via $K_S(\underline{x}, \underline{x}') = \underline{\phi}_S(\underline{x}) \cdot \underline{\phi}_S(\underline{x}')$. We require that $Z(\underline{x}_\mu)\hat{Z}_S(\underline{x}_\mu) > 1 - \xi_\mu$ for all training points. Ideally we want to have some large margins $1 - \xi_\mu = 1$, but we allow some of them to be smaller by introducing the *slack variables* $\xi_\mu$ and penalizing large values. To achieve this the following constrained minimization is performed:

$$\min_{\underline{w},b,\underline{\xi}} \frac{1}{2}\|\underline{w}\|^2 + C\sum_\mu \xi_\mu \quad \text{subjected to} \quad \forall \mu \; Z(\underline{x}_\mu)\left[\underline{w} \cdot \underline{\phi}_S(\underline{x}_\mu) + b\right] \geq 1 - \xi_\mu, \; \xi_\mu \geq 0. \tag{11}$$

This problem can be expressed in a dual formulation as

$$\min_{\underline{a}} \frac{1}{2}\underline{a} \cdot \mathbb{Q}_S\underline{a} - \sum_{\mu=1}^n a_\mu \quad \text{subjected to} \quad \underline{Z} \cdot \underline{a} = 0, \; 0 \leq a_\mu \leq C, \tag{12}$$

where $\mathbb{Q}_{S,\mu,\nu} = Z(\underline{x}_\mu)Z(\underline{x}_\nu)K_S(\underline{x}_\mu,\underline{x}_\nu)$ and $\underline{Z}$ is the vector of the labels of the training points. Here $C \, (= 10^4$ in our simulations) controls the trade-off between minimizing the training error and maximizing the *margins* $1 - \xi_\mu$. For the details we refer to (Scholkopf and Smola, 2001). If $\underline{a}^\star$ is the solution to the minimization problem, than

$$\underline{w}^\star = \sum_\mu a_\mu^\star \underline{\phi}_S(\underline{x}_\mu), \tag{13}$$

$$b^\star = Z(\underline{x}_\mu) - \sum_\nu a_\nu y_\nu K_S(\underline{x}_\mu,\underline{x}_\nu) \quad \text{for any } \mu \text{ such that } a_\mu < C. \tag{14}$$

The predicted label for unseen data points is then

$$\text{sign}(\hat{Z}_S(\underline{x})) = \text{sign}(\sum_\mu Z(\underline{x}_\mu)a_\mu K_S(\underline{x}_\mu,\underline{x}) + b^\star) \tag{15}$$

The generalization error is now defined as the probability that an unseen image has a predicted label different from the true one, and such a probability is again estimated as an average over a test set with $n_{\text{test}}$ elements:

$$\text{Error} = \frac{1}{n_{\text{test}}} \sum_{\mu=1}^{n_{\text{test}}} \theta\left[-\text{sign}\left(\hat{Z}_S(\underline{x}_\mu)\right)Z(\underline{x}_\mu)\right]. \tag{16}$$

## B  DIFFERENT KERNEL VARIANCES

In Fig. 4 we show the learning curves for kernel regression on the MNIST (parity) dataset — the same setting as in Fig. 1 *(a)*. Several Laplace kernels of varying variance $\sigma$ are used. The variance ranges several orders of magnitude and the learning curves all decay with the same exponent, although for $\sigma = 10$ the algorithm achieves suboptimal performance and the test errors are increased by some factor.

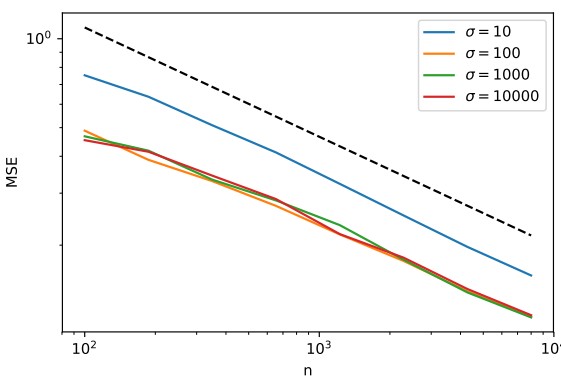

Figure 4: Learning curves for kernel regression on the MNIST dataset. Regression is performed with several Laplace kernels of varying variance $\sigma$ ranging from $\sigma = 10$ to $\sigma = 10000$.

## C  DIFFERENT CHOICE OF KERNEL VARIANCES

In Fig. 5 we show the learning curves for the Teacher-Student kernel regression problem, with a Student kernel that is always Laplace and a Teacher that can be either Gaussian or Laplace. We show how the test error decays with the size of the training dataset and how the asymptotic exponent depends on the spatial dimension. Every experiment is run with two different choices of the kernel variances: in one case $\sigma_T = \sigma_S = d$ and in the other $\sigma_T = \sigma_S = 10$. We observed that scaling the variances with the spatial dimension leads faster to the results that we predicted in this paper, but overall the choice has little effect on the exponents (both tend towards the prediction as the dataset size is increased).

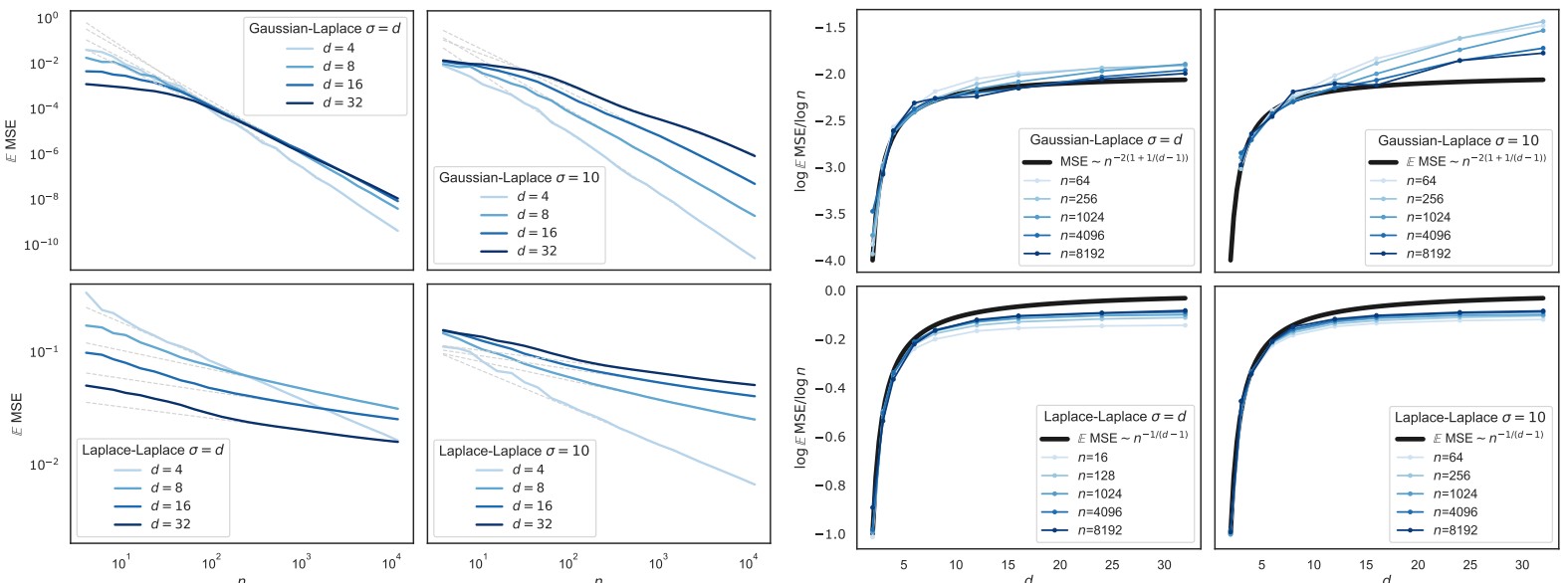

Figure 5: In these plots we show the results for the Teacher-Student kernel regression. The Student is always a Laplace kernel, the Teacher is either Gaussian or Laplace. The four plots on the left depict the mean-square error against the size of the dataset for different spatial dimensions of the data, those on the right show the fitted asymptotic exponent against the spatial dimension for different dataset sizes. For every case we show both the the results for $\sigma_T = \sigma_S = d$ and $\sigma_T = \sigma_S = 10$.

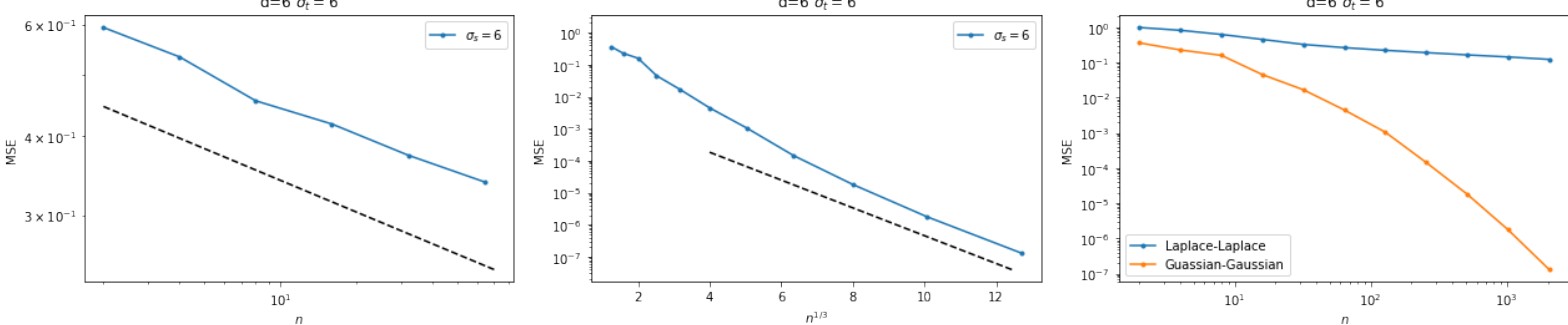

Figure 6: **Left:** The test error of a Laplace Teacher ($\alpha_T = d + 1$) with a Gaussian Student ($\alpha_S = \infty$) decays as a power law with the predicted exponent $\beta = \frac{1}{d}\min(1, \infty) = \frac{1}{6}$ in $d = 6$ dimensions. **Center:** When both the Teacher and the Student are Gaussian the test error decays faster than any power law as the number $n$ of data is increased. This plot confirm this by showing that the logarithm of the test error decays linearly as a function of $n^{\frac{1}{3}}$. **Right:** Comparison between the learning curves for the cases where both kernels are either Laplace (top blue line) or Gaussian (bottom orange line). While the former decays algebraically with the predicted exponent, the latter decays exponentially, in agreement with the prediction $\beta = \infty$ found within our framework. In all these plots we have taken the variances of both the Teacher and Student kernels to be equal to the dimension $d = 6$.

## D  GAUSSIAN STUDENTS

In this appendix we present the learning curves of Gaussian Students: the Fourier transform of these kernels decays faster than any power law and one can effectively consider $\alpha_S = \infty$. If the Teacher is Laplace ($\alpha_T = d + 1$) then the predicted exponent is finite and takes the values $\beta = \frac{1}{d}\min(\alpha_T - d, 2\alpha_S) = \frac{1}{d}\min(1, \infty) = \frac{1}{d}$. Such a case is displayed in Fig. 6 *(left)* in dimension $d = 6$. However, if we consider the Teacher to be Gaussian as well, then the predicted exponent would be $\beta = \frac{1}{d}\min(\infty, \infty) = \infty$. This case corresponds to Fig. 6 *(center)*: the test errors decays faster than a power law. In Fig. 6 *(right)* we compare the case where both kernels are Gaussian to the case where both kernels are Laplace: while the latter decays as a power law, the former decays much faster.

## E  MATÉRN TEACHERS

To further test the applicability of our theory, we show here some numerical simulations for a Teacher kernel that is a Matérn covariance function and a Laplace kernel as student. We ran the simulations in 1d: the data points are sampled uniformly on a 1-dimensional circle embedded in $\mathbb{R}^2$. Matérn kernels are parametrized by a parameter $\nu > 0$:

$$K_T(\underline{x}) = \frac{2^{1-\nu}}{\Gamma(\nu)} z^\nu \mathcal{K}_\nu(z), \tag{17}$$

where $z = \sqrt{2\nu}\frac{\|x\|}{\sigma}$ ($\sigma$ being the kernel variance), $\Gamma$ is the gamma function and $\mathcal{K}_\nu$ is the Bessel function of the second kind with parameter $\nu$. Interestingly we recover the Laplace kernel for $\nu = 1/2$ and the Gaussian kernel for $\nu = \infty$. As one can find in e.g. (Williams and Rasmussen, 2006), the exponent $\alpha_T$ that governs the decay at high frequency of this kernels is $\alpha_T = d + 2\nu$. Varying $\nu$ we can change the smoothness of the target function.

For $d = 1$ our prediction for the learning curve exponent $\beta$ is

$$\beta = \frac{1}{d}\min(\alpha_T - d, 2\alpha_S) = \min(2\nu, 4). \tag{18}$$

In Fig. 7 we verify that our prediction matches the numerical results.

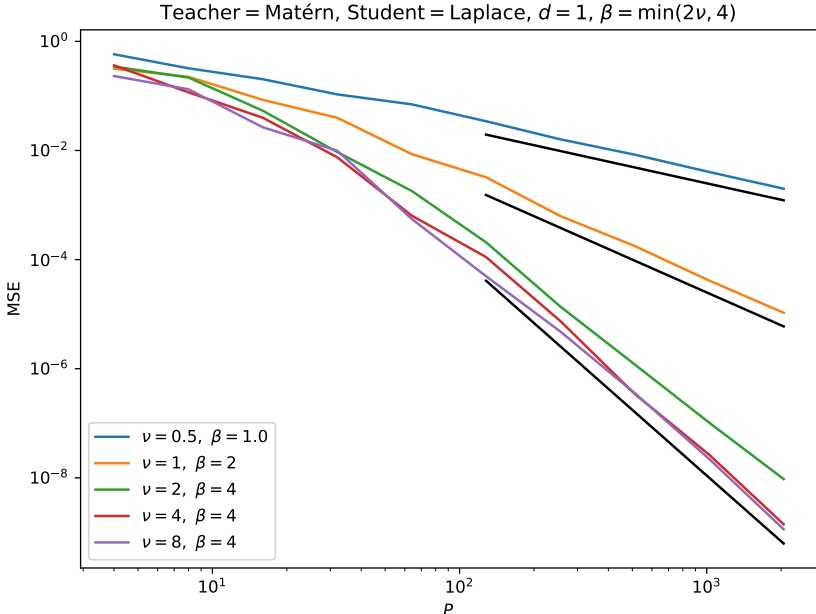

Figure 7: Mean-squared error for Matérn Teacher kernels and Laplace students. The variance of the kernels is equal to 2 for all the curves.

## F    PROOF OF THEOREM

We prove here Theorem 1:

**Theorem 1** *Let $\tilde{K}_T(\underline{w}) = c_T \|\underline{w}\|^{-\alpha_T} + o\left(\|\underline{w}\|^{-\alpha_T}\right)$ and $\tilde{K}_S(\underline{w}) = c_S \|\underline{w}\|^{-\alpha_S} + o\left(\|\underline{w}\|^{-\alpha_S}\right)$ as $\|\underline{w}\| \to \infty$, where $\tilde{K}_T(\underline{w})$ and $\tilde{K}_S(\underline{w})$ are the Fourier transforms of the kernels $K_T(\underline{x})$, $K_S(\underline{x})$ respectively, assumed to be positive definite. We assume $\tilde{K}_T(\underline{w})$ and $\tilde{K}_S(\underline{w})$ has a finite limit as $\|\underline{w}\| \to 0$ and that $K(\underline{0}) < \infty$. Then,*

$$\mathbb{E}\,\mathrm{MSE} = n^{-\beta} + o\left(n^{-\beta}\right) \quad \text{with} \quad \beta = \frac{1}{d}\min(\alpha_T - d, 2\alpha_S). \tag{19}$$

*Moreover, in the case of a Gaussian kernel the result holds valid if we take the corresponding exponent to be $\alpha = \infty$.*

*Proof.* Our strategy is to compute how the mean-square test error scales with distance $\delta$ between two nearest neighbors on the $d$-dimensional regular lattice. At the end, we will use the fact that $\delta \propto n^{-1/d}$, where $n$ is the number of sampled points on the lattice.

We denote by $\tilde{F}(\underline{w})$ the Fourier transform of a function $F : \mathcal{V} \to \mathbb{R}$:

$$\tilde{F}(\underline{w}) = L^{-d/2} \int_{\mathcal{V}} \mathrm{d}\underline{x}\, e^{-i\underline{w}\cdot\underline{x}} F(\underline{x}), \quad \text{where } \underline{w} \in \mathbb{L} \equiv \frac{2\pi}{L}\mathbb{Z}^d, \tag{20}$$

$$F(\underline{x}) = L^{-d/2} \sum_{\underline{w}\in\mathbb{L}} e^{i\underline{w}\cdot\underline{x}} \tilde{F}(\underline{w}). \tag{21}$$

If $Z \sim \mathcal{N}(0, K)$ is a Gaussian field with translation-invariant covariance $K$ then by definition

$$\mathbb{E} Z(\underline{x}) Z(\underline{x}') = K(\underline{x} - \underline{x}'). \tag{22}$$

*Properties of the Fourier transform of a Gaussian field:*

$$\tilde{K}(\underline{w}) = \tilde{K}(-\underline{w}) \in \mathbb{R}, \tag{23}$$

$$\mathbb{E}\tilde{Z}(\underline{w}) = 0, \tag{24}$$

$$\mathbb{E}\tilde{Z}(\underline{w})\overline{\tilde{Z}(\underline{w}')} = L^{d/2}\delta_{\underline{w}\underline{w}'}\tilde{K}(\underline{w}). \tag{25}$$

Eq. (23) comes from the fact that $K(\underline{x})$ is an even, real-valued function. The real and imaginary parts of $\tilde{Z}(\underline{w})$ are Gaussian random variables. They are all independent except that $\tilde{Z}(-\underline{w}) = \overline{\tilde{Z}(\underline{w})}$. Eq. (25) follows from the fact that $Z(\underline{x})$ and $K(\underline{x})$ are $L$-periodic functions, and therefore $e^{i\underline{w}\cdot\underline{x}}\tilde{K}(\underline{w})$ is the Fourier transform of $K(\cdot + \underline{x})$ if $\underline{w} \in \frac{2\pi}{L}\mathbb{Z}^d$. #

The solution Eq. (3) for kernel regression has two interpretations. In Section 4 we introduced it as the quantity that minimizes a quadratic error, but it can also be seen as the *maximum-a-posteriori* (MAP) estimation of another formulation of the problem (Williams and Rasmussen, 2006). The field $Z(\underline{x})$ is assumed to be drawn from a Gaussian distribution with covariance function $K_S(\underline{x})$: $K_S$ therefore plays a role in the *prior* distribution of the data $\underline{Z} = (Z(\underline{x}_\mu)_{\mu=1}^n)$. Inference about the value of the field $\hat{Z}_S(\underline{x})$ at another location is then performed by maximizing its posterior distribution,

$$\hat{Z}_S(\underline{x}) \equiv \arg\max \mathcal{P}\left(Z(\underline{x})|\underline{Z}\right). \tag{26}$$

Such a posterior distribution is Gaussian, and its mean — and therefore also the value that maximizes the probability — is exactly Eq. (3):

$$\hat{Z}_S(\underline{x}) = \underline{k}_S(\underline{x}) \cdot \mathbb{K}_S^{-1}\underline{Z}, \tag{27}$$

where where $\underline{Z} = \left(Z(\underline{x}_\mu)\right)_{\mu=1}^n$ are the training data, $\underline{k}_S(\underline{x}) = \left(K_S(\underline{x}_\mu, \underline{x})\right)_{\mu=1}^n$ and $\mathbb{K}_S = \left(K_S(\underline{x}_\mu, \underline{x}_\nu)\right)_{\mu,\nu=1}^n$ is the Gram matrix, that is invertible since the kernel $K_S$ is assumed to be positive definite. By Fourier transforming this relation we find

$$\tilde{Z}_S(\underline{w}) = \tilde{Z}^\star(\underline{w})\frac{\tilde{K}_S(\underline{w})}{\tilde{K}_S^\star(\underline{w})}, \tag{28}$$

where we have defined $F^\star(\underline{w}) \equiv \sum_{\underline{n}\in\mathbb{Z}^d} F\left(\underline{w} + \frac{2\pi\underline{n}}{\delta}\right)$ for a generic function $F$.

Another way to reach Eq. (28) is to consider that we are observing the quantities

$$\tilde{Z}^\star(\underline{w}) \equiv \delta^d L^{-d/2} \sum_{\underline{x}\in\text{lattice}} e^{-i\underline{w}\cdot\underline{x}}Z(\underline{x}) \equiv \sum_{\underline{n}\in\mathbb{Z}^d} \tilde{Z}\left(\underline{w} + \frac{2\pi\underline{n}}{\delta}\right). \tag{29}$$

Given that we know the prior distribution of the Fourier components on the right-hand side in Eq. (29), we can infer their posterior distribution once their sums are constrained by the value of $\tilde{Z}^\star(\underline{w})$, and it is straightforward to see that we recover Eq. (28).

The mean-square error can then we written using the Parseval-Plancherel identity,

$$\mathbb{E}\,\text{MSE} = L^{-d}\mathbb{E}\int_{\mathcal{V}} d\underline{x}\,[Z(\underline{x}) - \hat{Z}_S(\underline{x})]^2 = L^{-d}\mathbb{E}\sum_{\underline{w}\in\mathbb{L}}\left|\tilde{Z}(\underline{w}) - \tilde{Z}^\star(\underline{w})\frac{\tilde{K}_S(\underline{w})}{\tilde{K}_S^\star(\underline{w})}\right|^2. \tag{30}$$

By taking the expectation value with respect to the Teacher and using Eq. (23)-Eq. (25) we can write the mean-square error as

$$
\mathbb{E}\,\mathrm{MSE} = L^{-d}\mathbb{E}\sum_{\underline{w}\in\mathbb{L}}\left[\tilde{Z}(\underline{w})\overline{\tilde{Z}(\underline{w})} - 2\tilde{Z}(\underline{w})\overline{\tilde{Z}^{\star}(\underline{w})}\frac{\tilde{K}_S(\underline{w})}{\tilde{K}_S^{\star}(\underline{w})} + \tilde{Z}^{\star}(\underline{w})\overline{\tilde{Z}^{\star}(\underline{w})}\frac{\tilde{K}_S^2(\underline{w})}{\tilde{K}_S^{\star^2}(\underline{w})}\right] =
$$

$$
= L^{-d}\mathbb{E}\sum_{\underline{w}\in\mathbb{L}}\tilde{Z}(\underline{w})\overline{\tilde{Z}(\underline{w})} - 2\frac{\tilde{K}_S(\underline{w})}{\tilde{K}_S^{\star}(\underline{w})}\sum_{\underline{n}\in\mathbb{Z}^d}\tilde{Z}(\underline{w})\overline{\tilde{Z}\left(\underline{w}+\frac{2\pi\underline{n}}{\delta}\right)} +
$$

$$
+ \frac{\tilde{K}_S^2(\underline{w})}{\tilde{K}_S^{\star^2}(\underline{w})}\sum_{\underline{n},\underline{n}'\in\mathbb{Z}^d}\tilde{Z}\left(\underline{w}+\frac{2\pi\underline{n}}{\delta}\right)\overline{\tilde{Z}\left(\underline{w}+\frac{2\pi\underline{n}'}{\delta}\right)} =
$$

$$
= L^{-d/2}\sum_{\underline{w}\in\mathbb{L}}\tilde{K}_T(\underline{w}) - 2\frac{\tilde{K}_S(\underline{w})}{\tilde{K}_S^{\star}(\underline{w})}\tilde{K}_T(\underline{w}) + \frac{\tilde{K}_S^2(\underline{w})}{\tilde{K}_S^{\star^2}(\underline{w})}\sum_{\underline{n}\in\mathbb{Z}^d}\tilde{K}_T\left(\underline{w}+\frac{2\pi\underline{n}}{\delta}\right) =
$$

$$
= L^{-d/2}\sum_{\underline{w}\in\mathbb{L}\cap\mathcal{B}}\tilde{K}_T^{\star}(\underline{w}) - 2\frac{[\tilde{K}_T\tilde{K}_S]^{\star}(\underline{w})}{\tilde{K}_S^{\star}(\underline{w})} + \frac{\tilde{K}_T^{\star}(\underline{w})[\tilde{K}_S^2]^{\star}(\underline{w})}{\tilde{K}_S^{\star}(\underline{w})^2}, \quad (31)
$$

where $\mathcal{B} = \left[-\frac{\pi}{\delta}, \frac{\pi}{\delta}\right]^d$ is the Brillouin zone.

At high frequencies, $\tilde{K}_T(\underline{w}) = c_T\|\underline{w}\|^{-\alpha_T} + o\left(\|\underline{w}\|^{-\alpha_T}\right)$ and $\tilde{K}_S(\underline{w}) = c_S\|\underline{w}\|^{-\alpha_S} + o\left(\|\underline{w}\|^{-\alpha_S}\right)$. Therefore:

$$
\tilde{K}_T^{\star}(\underline{w}) = \tilde{K}_T(\underline{w}) + \delta^{\alpha_T}c_T\sum_{\underline{n}\in\mathbb{Z}^d\setminus\{\underline{0}\}}\|\underline{w}\delta + 2\pi\underline{n}\|^{-\alpha_T} + o\left(\|\underline{w}\|^{-\alpha_T}\right) \equiv
$$

$$
\equiv \tilde{K}_T(\underline{w}) + \delta^{\alpha_T}c_T\,\psi_T(\underline{w}\delta) + o\left(\|\underline{w}\|^{-\alpha_T}\right). \quad (32)
$$

This equation defines the function $\psi_T$, and a similar equation holds for the Student as well. The hypothesis $K_T(\underline{0}) \propto \int \mathrm{d}\underline{w}\,\tilde{K}_T(\underline{w}) < \infty$ implies $\alpha_T > d$ and therefore $\sum_{\underline{n}\in\mathbb{Z}^d}\|\underline{n}\|^{-\alpha_T} < \infty$ (and likewise for the Student). Then, $\psi_{\alpha_T}(\underline{0}), \psi_{\alpha_S}(\underline{0})$ are finite; furthermore, the $\underline{w}$'s in the sum Eq. (31) are at most of order $\mathcal{O}\left(\delta^{-1}\right)$, therefore the terms $\psi_\alpha(\underline{w}\delta)$ are $\mathcal{O}(\delta^0)$ and do not influence how Eq. (31) scales with $\delta$. Applying Eq. (32), expanding for $\delta \ll 1$ and keeping only the leading orders, we find

$$
\mathbb{E}\,\mathrm{MSE} =
$$

$$
= L^{-d/2}\left[\sum_{\underline{w}\in\mathbb{L}\cap\mathcal{B}}2c_T\psi_{\alpha_T}(\underline{w}\delta)\delta^{\alpha_T} + c_S^2\psi_{2\alpha_S}(\underline{w}\delta)\frac{\tilde{K}_T(\underline{w})}{\tilde{K}_S^2(\underline{w})}\delta^{2\alpha_S} + o\left(\|\underline{w}\|^{-\alpha_T}\right) + o\left(\|\underline{w}\|^{-2\alpha_S}\right)\right] =
$$

$$
= L^{-d/2}\left[\sum_{\underline{w}\in\mathbb{L}\cap\mathcal{B}}2c_T\psi_{\alpha_T}(\underline{w}\delta)\delta^{\alpha_T} + c_S^2\psi_{2\alpha_S}(\underline{w}\delta)\frac{\tilde{K}_T(\underline{w})}{\tilde{K}_S^2(\underline{w})}\delta^{2\alpha_S}\right] + o\left(\|\underline{w}\|^{-\alpha_T-d}\right) + o\left(\|\underline{w}\|^{-2\alpha_S-d}\right).
$$

$$
(33)
$$

We have neglected terms proportional to, for instance, $\delta^{\alpha_T+\alpha_S}$, since they are subleading with respect to $\delta^{\alpha_T}$, but we must keep both $\delta^{\alpha_T}$ and $\delta^{\alpha_S}$ since we do not know a priori which one is dominant. The additional term $\delta^{-d}$ in the subleading terms comes from the fact that $|\mathbb{L}\cap\mathcal{B}| = \mathcal{O}\left(\delta^{-d}\right)$.

The first term in Eq. (33) is the simplest to deal with: since $\|\underline{w}\delta\|$ is smaller than some constant for all $\underline{w} \in \mathbb{L}\cap\mathcal{B}$ and the function $\psi_{\alpha_T}(\underline{w}\delta)$ has a finite limit, we have

$$
\delta^{\alpha_T}\sum_{\underline{w}\in\mathbb{L}\cap\mathcal{B}}2c_T\psi_{\alpha_T}(\underline{w}\delta) = \mathcal{O}\left(\delta^{\alpha_T}|\mathbb{L}\cap\mathcal{B}|\right) = \mathcal{O}\left(\delta^{\alpha_T-d}\right). \quad (34)
$$

We then split the second term in Eq. (33) in two contributions:

**Small** $\|\underline{w}\|$  We consider "small" all the terms $\underline{w} \in \mathbb{L} \cap \mathcal{B}$ such that $\|\underline{w}\| < \Gamma$, where $\Gamma \gg 1$ is $\mathcal{O}(\delta^0)$ but large. As $\delta \to 0$, $\psi_{2\alpha_S}(\underline{w}\delta) \to \psi_{2\alpha_S}(\underline{0})$ which is finite because $K(\underline{0}) < \infty$. Therefore

$$\delta^{2\alpha_S} \sum_{\substack{\underline{w} \in \mathbb{L} \cap \mathcal{B} \\ \|\underline{w}\| < \Gamma}} c_S^2 \psi_{2\alpha_S}(\underline{w}\delta) \frac{\tilde{K}_T(\underline{w})}{\tilde{K}_S^2(\underline{w})} \to \delta^{2\alpha_S} c_S^2 \psi_{2\alpha_S}(\underline{0}) \sum_{\substack{\underline{w} \in \mathbb{L} \cap \mathcal{B} \\ \|\underline{w}\| < \Gamma}} \frac{\tilde{K}_T(\underline{w})}{\tilde{K}_S^2(\underline{w})}. \tag{35}$$

The summand is real and strictly positive because the positive definiteness of the kernels implies that their Fourier transforms are strictly positive. Moreover, as $\delta \to 0$, $\mathbb{L} \cap \mathcal{B} \cap \{\|\underline{w}\| < \Gamma\} \to \mathbb{L} \cap \{\|\underline{w}\| < \Gamma\}$, which contains a finite number of elements, independent of $\delta$. Therefore

$$\delta^{2\alpha_S} \sum_{\substack{\underline{w} \in \mathbb{L} \cap \mathcal{B} \\ \|\underline{w}\| < \Gamma}} c_S^2 \psi_{2\alpha_S}(\underline{w}\delta) \frac{\tilde{K}_T(\underline{w})}{\tilde{K}_S^2(\underline{w})} = \mathcal{O}\left(\delta^{2\alpha_S}\right). \tag{36}$$

**Large** $\|\underline{w}\|$  "Large" $\underline{w}$ are those with $\|\underline{w}\| > \Gamma$: we recall that $\Gamma \gg 1$ is $\mathcal{O}(\delta^0)$ but large. This allows us to approximate $\tilde{K}_T$, $\tilde{K}_S$ in the sum with their asymptotic behavior:

$$\delta^{2\alpha_S} \sum_{\substack{\underline{w} \in \mathbb{L} \cap \mathcal{B} \\ \|\underline{w}\| > \Gamma}} c_S^2 \psi_{2\alpha_S}(\underline{w}\delta) \frac{\tilde{K}_T(\underline{w})}{\tilde{K}_S^2(\underline{w})} \propto \delta^{2\alpha_S} \sum_{\substack{\underline{w} \in \mathbb{L} \cap \mathcal{B} \\ \|\underline{w}\| > \Gamma}} \|\underline{w}\|^{-\alpha_T + 2\alpha_S} + o\left(\|\underline{w}\|^{-\alpha_T + 2\alpha_S}\right) \approx$$

$$\approx \delta^{2\alpha_S} \int_\Gamma^{1/\delta} \mathrm{d}w \, w^{d-1-\alpha_T + 2\alpha_S} + o\left(\|\underline{w}\|^{-\alpha_T + 2\alpha_S}\right) = \mathcal{O}\left(\delta^{\min(\alpha_T - d, 2\alpha_S)}\right). \tag{37}$$

Finally, putting Eq. (34), Eq. (36) and Eq. (37) together,

$$\mathbb{E}\,\mathrm{MSE} = \mathcal{O}\left(\delta^{\min(\alpha_T - d, 2\alpha_S)}\right). \tag{38}$$

The proof is concluded by considering that $\delta = \mathcal{O}\left(n^{-1/d}\right)$.

In the case of a Gaussian kernel $K(\underline{x}) \propto \exp(-\|\underline{x}\|^2/(2\sigma^2))$ — and therefore $\tilde{K}(\underline{w}) \propto \exp(-\sigma^2\|\underline{w}\|^2/2)$ — one has to redo the calculations starting from Eq. (31), but the final result can be easily recovered by taking the limit $\alpha \to +\infty$ (Gaussian kernels decay faster than any power law).

$\square$

## G   PROOFS OF LEMMAS

**Lemma 1** *Let $K(\underline{x}, \underline{x}')$ be a translation-invariant isotropic kernel such that $\tilde{K}(\underline{w}) = c\|\underline{w}\|^{-\alpha} + o\left(\|\underline{w}\|^{-\alpha}\right)$ as $\|\underline{w}\| \to \infty$ and $\|\underline{w}\|^d \tilde{K}(\underline{w}) \to 0$ as $\|w\| \to 0$. If $\alpha > d + n$ for some $n \in \mathbb{Z}^+$, then $K(\underline{x}) \in C^n$, that is, it is at least $n$-times differentiable.*

*Proof.* The kernel is rotational invariant in real space ($K(\underline{x}) = K(\|\underline{x}\|)$) and therefore also in the frequency domain. Then, calling $\hat{\epsilon}_1 = (1, 0, \dots)$ the unitary vector along the first dimension $x_1$,

$$K(x) \propto \int \mathrm{d}\underline{w} \, e^{i\underline{w} \cdot \hat{\epsilon}_1 x} \tilde{K}(\|\underline{w}\|). \tag{39}$$

It follows that

$$|\partial^m K(x)| \propto \left| \int \mathrm{d}\underline{w} \, (\underline{w} \cdot \hat{\epsilon}_1)^m e^{i\underline{w} \cdot \hat{\epsilon}_1 x} \tilde{K}(\|\underline{w}\|) \right| < \int \mathrm{d}\underline{w} \, |\underline{w} \cdot \hat{\epsilon}_1|^m |\tilde{K}(\|\underline{w}\|)| \propto$$

$$\propto \int_0^\infty \mathrm{d}w \, w^{d-1+m} |\tilde{K}(w)| \int_0^\pi \mathrm{d}\phi_1 |\cos(\phi_1)|^m \propto \int_0^\infty \mathrm{d}w \, w^{d-1+m} |\tilde{K}(w)|. \tag{40}$$

We want to claim that this quantity is finite if $m \le n$. Convergence at infinity requires $m < \alpha - d$, that is always smaller than or equal to $n$ because of the hypothesis of the lemma. Convergence in zero requires that $w^{d+m} |\tilde{K}(w)| \to 0$, and we want this to hold for all $0 \le m < \alpha - d$, the most constraining one being the condition with $m = 0$. $\square$

**Lemma 2** *Let $Z \sim \mathcal{N}(0, K)$ be a $d$-dimensional Gaussian random field, with $K \in C^{2n}$ being a $2n$-times differentiable kernel. Then $Z$ is $n$-times differentiable in the sense that*

- *derivatives of $Z(\underline{x})$ are a Gaussian random fields;*
- $\mathbb{E}\partial_{x_1}^{n_1} \cdots \partial_{x_d}^{n_d} Z(\underline{x}) = 0$;
- $\mathbb{E}\partial_{x_1}^{n_1} \cdots \partial_{x_d}^{n_d} Z(\underline{x}) \cdot \partial_{x_1}^{n'_1} \cdots \partial_{x_d}^{n'_d} Z(\underline{x}') = \partial_{x_1}^{n_1+n'_1} \cdots \partial_{x_d}^{n_d+n'_d} K(\underline{x} - \underline{x}') < \infty$ *if the derivatives of $K$ exist.*

*In particular, $\mathbb{E}\partial_{x_i}^m Z(\underline{x}) \cdot \partial_{x_i}^m Z(\underline{x}') = \partial_{x_i}^{2m} K(\underline{x} - \underline{x}') < \infty \ \forall m \leq n$.*

*Proof.* Derivatives of $Z(\underline{x})$ are defined as limits of sums and differences of the field $Z$ evaluated at different points, therefore they are Gaussian random fields too, and furthermore it is straightforward to see that their expected value is always $0$ if the field itself is zero centered.

The correlation can be computed via induction. Assume that $\mathbb{E}\partial_{x_1}^{n_1} \cdots \partial_{x_d}^{n_d} Z(\underline{x}) \cdot \partial_{x_1}^{n'_1} \cdots \partial_{x_d}^{n'_d} Z(\underline{x}') = \partial_{x_1}^{n_1+n'_1} \cdots \partial_{x_d}^{n_d+n'_d} K(\underline{x} - \underline{x}')$ holds true. Then, if we increment $n_1$:

$$
\mathbb{E}\partial_{x_1}^{n_1+1} \cdots \partial_{x_d}^{n_d} Z(\underline{x}) \cdot \partial_{x_1}^{n'_1} \cdots \partial_{x_d}^{n'_d} Z(\underline{x}') =
$$
$$
= \lim_{h \to 0} h^{-1} \mathbb{E}\left[ \partial_{x_1}^{n_1} \cdots \partial_{x_d}^{n_d} Z(\underline{x} + h\hat{\epsilon}_1) - \partial_{x_1}^{n_1} \cdots \partial_{x_d}^{n_d} Z(\underline{x}) \right] \cdot \partial_{x_1}^{n'_1} \cdots \partial_{x_d}^{n'_d} Z(\underline{x}') =
$$
$$
= \lim_{h \to 0} h^{-1} \left[ \partial_{x_1}^{n_1+n'_1} \cdots \partial_{x_d}^{n_d+n'_d} K(\underline{x} - \underline{x}' + h\hat{\epsilon}_1) - \partial_{x_1}^{n_1+n'_1} \cdots \partial_{x_d}^{n_d+n'_d} K(\underline{x} - \underline{x}') \right] =
$$
$$
= \partial_{x_1}^{n_1+1+n'_1} \cdots \partial_{x_d}^{n_d+n'_d} K(\underline{x} - \underline{x}'). \quad (41)
$$

Of course by symmetry the same can be said about the increase of any other exponent. To conclude the induction proof we simply recall that by definition $\mathbb{E}Z(\underline{x})Z(\underline{x}') = K(\underline{x} - \underline{x}')$. $\qquad \square$

