# OpenReview forum: "Asymptotic learning curves of kernel methods: empirical data v.s. Teacher-Student paradigm"
_ICLR.cc/2020/Conference — Reject_

### Official Review · AnonReviewer2 · 2019-10-23
**Official Blind Review #2**

**Rating:** 3

**Review:**

This paper experimentally investigates how fast the generalization error decreases when some specific kernel functions are used in real datasets. This paper conducted numerical experiments on several datasets to investigate the decreasing rate of the generalization error, and the rate is determined for such datasets. This decreasing rate is theoretically analyzed by using the approximation theory of RKHS in the teacher-student setting. It is shown that the rate is determined with the smoothness and effective dimensionality of input. Then, the smoothness of the teacher function is also derived through this analysis.

Overall, the paper is well written. I could easily follow the line. The pros and cons of the paper are summarized as follows.

Pros:
The numerical experimetns conducted in this paper are thorough, and they show interesting observations on the real datasets. This paper gives a practical information on the theoretical analysis as an empirical study.

Cons:
- The approximation theory shown in this paper (Theorem 1) is closely related to well-known results on kernel interpolation. However, this paper misses several related work in the literature. The result should be properly put in the literature. See, for example, [R1].

[R1] H. Wendland. Scattered Data Approximation. Cambridge University Press, Cambridge, UK, 2005.

- It is mentioned that this paper investigates the "generalization error." However, what is acutally done is more like "approximation error" analysis (about linear interpolation in RKHS). In reality, there are observation noises and thus we typically consider the generalization error. But, the teacher-student setting does not assume the existence of noise. Under existence of noise, generalization error analysis seems more appropriate as performed in [R2].

[R2] I. Steinwart and A. Christmann. Support Vector Machines. Springer, 2008.

Minor comment:
- In the introduction, it is mentioned that the assumption that the target function is included in RKHS is strong. However, the teacher-student setting considered in Theorem 1 assumes this assumption. The introduction requires some modification to make the message consistent.

---Update---
Thank you for your reply.
I understand the RKHS for teacher and that for student are different. But, in the introduction, you stated as "Yet, RKHS is a very strong assumption which requires the smoothness of the target function to increase with d (Bach, 2017) (see more on this point below), which may not be realistic in large dimensions.", which sounds like that an assumption that the target function is included in "some" RKHS corresponding to a smooth kernel is a strong assumption. At least, this sentence is not saying anything about difference between teacher and student, but is just saying assuming smoothness on the target is unrealistic. For me, this sounds inconsistent to your analysis. (This is just a minor concern. I wanted to clarify my understanding of your problem setting.)

I think the setting where the teacher is not included in the student RKHS is also analyzed, for example, in the following papers (there are also several related papers):
F.J. Narcowich, J.D. Ward, and H. Wendland. Sobolev Error Estimates and a Bernstein
Inequality for Scattered Data Interpolation via Radial Basis Functions. Constr. Approx.,
24:175–186, 2006.
SCHEUERER, M., SCHABACK, R., & SCHLATHER, M. (2013). Interpolation of spatial data – A stochastic or a deterministic problem? European Journal of Applied Mathematics, 24(4), 601-629.

Therefore, I still feel that the paper requires more expositions about the relation to the literature.

**Experience Assessment:**

I have read many papers in this area.

**Review Assessment: Checking Correctness Of Derivations And Theory:**

I assessed the sensibility of the derivations and theory.

**Review Assessment: Checking Correctness Of Experiments:**

I assessed the sensibility of the experiments.

**Review Assessment: Thoroughness In Paper Reading:**

I read the paper at least twice and used my best judgement in assessing the paper.

---

> ### Author Response · Authors · 2019-11-11
> **Answer to R2**
>
> We thank R2 for pointing to  literature. However reading (in the short time we had) the book by Wendland we could only find cases where the target function is assumed to be in the RKHS of the kernel used to make the inference. This is definitely *not* what we do in our paper: our assumption is much weaker (a Gaussian process is never in the RKHS of its co-variance kernel), and leads to training curves with new exponents.  If R2 has references that correspond to what we actually do, we would be interested to know.
>
> As stated in (6) and (28) we are computing the test error, defined as the expected mean-squared error committed on a new, previously unobserved point $\mathbf{x}$. Of course in the presence of noise one could decompose the generalization error over different contributions. But again, the treatment of noise does not exist in the framework we introduce here, which is not based on RKHS.
>
> Concerning the last comment:
> This is wrong. The Teacher-Student setting considered in Theorem 1 precisely does *not* assume that the instance $Z$ of the teacher process lies in the RKHS of the student kernel, as also R1 has emphasized. A brief discussion of what would happen in that case is included in the Conclusion (Sec. 6), and it leads to the conclusion that in such a case $\alpha_T$ would have to be fairly large and $Z$ should be very smooth (in a mean-squared sense). All the previous comments of the referee appear to be based on this misconception, apparently leading to the weak mark he gave.

---

### Official Review · AnonReviewer1 · 2019-10-23
**Official Blind Review #1**

**Rating:** 6

**Review:**

This paper studies, empirically and theoretically, the learning rates of (shift-invariant) kernel learners in a misspecified setting. In the well-specified setting, the rate of kernel learners is at least $n^{-1/2}$, and in a misspecified setting assuming only Lipschitz targets, the rate is $n^{-1/d}$. Neither seems to match the experimental rate on MNIST and CIFAR-10; this paper proposes a theoretical model that can more-or-less match the experimental rate with essentially-reasonable assumptions.

My main complaint is on the basic setting of the work: in your motivation, you say "it is nowadays part of the lore that there exist kernels whose performance is nearly comparable to deep networks." The main such kernel, though, is the (convolutional) neural tangent kernel of Arora et al. (2019), which unlike the kernels you study here is not shift-invariant, and your theorems do not at all apply to this kernel. This is fine, but should probably be clearer in the description.

A related comment on your main theorem: your target function evaluated at every conceivable point (not just on a grid) is a sample from a Gaussian process. Samples from GPs with mean zero and covariance kernel $K_T$ almost surely are not in the RKHS $\mathcal H_T$, but they *are* almost surely in the RKHS of any kernel $K_R$ which nuclearly dominates $K_T$ (see Lukic and Beder, "Stochastic Processes with Sample Paths in Reproducing Kernel Hilbert Spaces", Trans. AMS 2001). If such a kernel exists, using it as the "student" kernel should give us a rate of at least $n^{-1/2}$ with standard results (with some slight details still to be worked out, but should be true). Thus, it seems that your theorem implies that for $\alpha_T < \frac32 d$, no such translation-invariant kernel $R$ exists. This might be already easy to see from a Fourier definition of nuclear dominance, I'm not sure, but if not it is something that seems of somewhat independent interest.

It is also notable that both your practical results and your theorem are for algorithms essentially without any regularization other than the choice of kernel: the regression setting is exact interpolation, and your soft-margin uses $C = 10^4$ so is "almost" a hard-margin SVM. This is also fine – interpolation methods have seen a lot of interest of late, and certainly can perform well. But it's not the typical setting, and it would be interesting to see if the curves of Figure 1 look different when using e.g. a cross-validated setting for the amount of regularization.

Another complaint: you argue that applying Theorem 1 with this particular notion of effective dimension seems to give good results, but at least as it's stated, Theorem 1 doesn't actually apply with effective dimension, only ambient dimension. Is it possible to prove Theorem 1 with an appropriate version of effective dimension? I didn't carefully check the proof, but from your outlined sketch it seems like it might be only a small change.

Empirically, your investigations are nice, but it would be good to consider some other shift-invariant kernels as well: inverse multiquadric, Matérn, or spline RBF kernels would be prominent options.

Overall: I think this is a worthwhile study with interesting results. The theoretical setting, though, is somewhat limited by its fundamental approach, and the experiments aren't as thorough as they could be. Also, honestly, I'm not sure ICLR is the best venue for it (if I had written this paper around this time, I probably would have submitted it to AISTATS; it's certainly not *off* topic for ICLR, but fairly distant from most work at it).

Some typos:
- Under (2): "man-square error."
- Under (25): "where where."

**Experience Assessment:**

I have published one or two papers in this area.

**Review Assessment: Checking Correctness Of Derivations And Theory:**

I assessed the sensibility of the derivations and theory.

**Review Assessment: Checking Correctness Of Experiments:**

I assessed the sensibility of the experiments.

**Review Assessment: Thoroughness In Paper Reading:**

I read the paper at least twice and used my best judgement in assessing the paper.

---

> ### Author Response · Authors · 2019-11-11
> **Answer to R1**
>
> We agree with R1: our point is that the analogy between deep learning and  kernels motivates a better understanding of kernels in general. We will clarify this sentence.
>
> R1 is correct, and it is a interesting statement.  We will add a few sentences introducing the notion of kernel dominance and stating this result.
>
> Studying how a regularizing term would affect the learning curve is an interesting empirical question. Yet we do not think it is opportune to add such studies here: they do not connect to our theoretical framework that does not include regularization. Our manuscript would then be less clear.
>
> Yes, the proof is identical in the case where the points lie on a regular lattice of lower dimension $d_\mathrm{eff}$ than the embedding space. To see that, it is sufficient to define the kernels restricted to this lower dimension subspace. The restricted kernels have the same coefficient $\alpha_S$ and $\alpha_T$; and the theorem goes through with $d$ replaced by $d_\mathrm{eff}$. We will indicate that point.
>
> Thanks! Concerning other kernels: note that our goal is not to perform a test of our theorem on all translation invariant kernels. Instead, it is to test all the qualitatively distinct predictions  our theorem makes. To do that we change the spatial dimension as well as the smoothness of the kernel (Laplace or Gaussian) both for the teacher and the student kernels. That way, we explore all  the different cases that our theorem predicts, and we believe the empirical support for our prediction is strong. However, should R1 still deem necessary that we provide some further numerical results, we will.

---

> > ### Comment · AnonReviewer1 · 2019-11-15
> > **Thanks**
> >
> > Thanks for your comments; I in particular think making explicit that the theorem applies with effective dimension is important.
> >
> > In terms of testing the theorem's predictions: I agree that you certainly do not need to study some enormous band of kernels. But you've only evaluated two smoothness settings (Laplace and Gaussian); it would bring more confidence to consider more. Matérn kernels in particular should allow for easily setting the smoothness parameter to whatever you want. I don't know if I would call this "necessary," but I think it would help illustrate the applicability of your theorem.

---

> > > ### Author Response · Authors · 2019-11-15
> > > **Further data**
> > >
> > > Thank you. Following your suggestion we ran some simulations using a Matérn kernel of varying parameter $\nu$ as Teacher and a Laplace kernel as student, in 1d.
> > >
> > > As found in [1] the exponent $\alpha$ for a Matérn kernel with parameter $\nu$ is $\alpha=d+2\nu$. Varying $\nu$ we can vary the mean-squared smoothness of the data. Within our framework we can predict the exponent $\beta$ to be $\beta = \frac1d \min(\alpha_T-d,2\alpha_S) = \min(2\nu,4)$. In the simulations we tested several values $\nu = 0.5, 1, 2, 4, 8$. Indeed, we observe the predicted exponents $\beta=1$ for $\nu=0.5$, $\beta=2$ for $\nu=1$ and $\beta=4$ for the others. We will definitely add this data to our paper, since we agree with you that they strengthen our point.
> > >
> > > Is there any way to upload the figure here for the review process?
> > >
> > >
> > > [1] Rasmussen, Carl Edward and Williams, Christopher K. I. (2006) Gaussian Processes for Machine Learning

---

> > > > ### Author Response · Authors · 2019-11-15
> > > > **Data added to pdf**
> > > >
> > > > We have added the new data in Appendix E in the pdf.

---

### Official Review · AnonReviewer3 · 2019-10-29
**Official Blind Review #3**

**Rating:** 3

**Review:**

In order to rationalize the existence of non-trivial exponents that can be independent of the specific kernel
used, this paper introduces the Teacher-Student framework for kernels. In this scheme, a Teacher generates data according to a Gaussian random field, and a Student learns them via kernel regression. Theresults quantify how smooth Gaussian data should be to avoid the curse of dimensionality, and indicate that for kernel learning the relevant dimension of the data should be defined in terms of how the distance between nearest data points depends on sample numbers.
The paper is well written, tghe major issue of this paper is the lack of comparison with other previous methods. Therefore, the efficacy of the proposed model can not be well demontrated.

**Experience Assessment:**

I do not know much about this area.

**Review Assessment: Checking Correctness Of Derivations And Theory:**

I assessed the sensibility of the derivations and theory.

**Review Assessment: Checking Correctness Of Experiments:**

I carefully checked the experiments.

**Review Assessment: Thoroughness In Paper Reading:**

I read the paper at least twice and used my best judgement in assessing the paper.

---

> ### Author Response · Authors · 2019-11-11
> **Answer to R3**
>
> Our manuscript does not provide a method (as noted by the other reviewers), so there is no meaning in comparing its efficacy to anything else.  It is a fundamental work, proposing a theoretical framework to explain quantitative observations on the learning curves of kernels.

---

### Decision · Program_Chairs · 2019-12-19

**Decision:**

Reject

**Comment:**

The paper studies, theoretically and empirically, the problem when generalization error decreases as $n^{-\beta}$ where $\beta$ is not $\frac{1}{2}$. It analyses a Teacher-Student problem where the Teacher generates data from a Gaussian random field. The paper provides a theorem that derives $\beta$ for Gaussian and Laplace kernels, and show empirical evidence supporting the theory using MNIST and CIFAR.

The reviews contained two low scores, both of which were not confident. A more confident reviewer provided a weak accept score, and interacted multiple times with the authors during the discussion period (which is one of the nice things about the ICLR review process). However, this reviewer also noted that ICLR may not be the best venue for this work.

Overall, while this paper shows promise, the negative review scores show that the topic may not be the best fit to the ICLR audience.